# The Role of *ERG11* Point Mutations in the Resistance of *Candida albicans* to Fluconazole in the Presence of Lactate

**DOI:** 10.3390/pathogens11111289

**Published:** 2022-11-03

**Authors:** Aneta K. Urbanek, Zofia Łapińska, Daria Derkacz, Anna Krasowska

**Affiliations:** 1Faculty of Biotechnology, University of Wroclaw, 50-383 Wroclaw, Poland; 2Department of Molecular and Cellular Biology, Faculty of Pharmacy, Wroclaw Medical University, Borowska 211, 50-556 Wroclaw, Poland

**Keywords:** *Candida albicans*, fluconazole (FLC), lactate, *ERG11* gene, point mutations

## Abstract

Candidiasis refers to both superficial and deep-tissue fungal infections often caused by *Candida albicans*. The treatment of choice for these infections is the use of azoles, such as fluconazole (FLC). However, the increased use of antifungal agents has led to the emergence of azole-resistant isolates of *C. albicans*. Thus, the development of alternative drugs that are more efficient and with a better toxicological profile is necessary. This study aimed to determine the susceptibility profile of *C. albicans* CAF2-1 strain to FLC in the presence of glucose or lactate. The research was also focused on single nucleotide polymorphism (SNP) and the determination of the effect of the identified point mutations on the amino acid sequence of the Erg11 protein. The results show the growth of *C. albicans* CAF2-1 in the presence of FLC was significantly lower in the presence of lactate than in glucose. As a result, among recorded 45 amino acid mutations, the following mutations may be associated with the reduced susceptibility of *C. albicans* to FLC: G10D, G10V, I11M, I11R, Y13N, F31V, L35F, A249D, Q250H, E266G, R267G, N273K, D275C, D275G, D275R. Moreover, a twice higher number of hot-spot mutations was found in the presence of glucose as a sole carbon source compared to cells grown on lactate.

## 1. Introduction

*Candida albicans* is an important opportunistic yeast that can cause various infections in humans, including superficial infections, such as oral or vaginal diseases, and life-threatening systemic infections [1]. Candidal infections are called candidiasis and have emerged as important public health problems. In particular, invasive candidiasis is associated with high rates of morbidity and mortality in humans [2,3].

The treatment of candidiasis is commonly associated with antifungal usage. *Candida* infections are treated with azole drugs (e.g., clotrimazole, miconazole, fluconazole (FLC)), polyenes (e.g., nystatin and amphotericin B), echinocandins (e.g., caspofungin, micafungin), nucleoside analogues and allylamines [2]. Recently, ibrexafungerp, the newest antifungal drug, has been approved for the treatment of vulvovaginal candidiasis [4,5]. Among the available antifungal agents, the preferred and most frequently used drugs in the treatment of *Candida* infections are azoles [6]. Azoles inhibit ergosterol biosynthesis by interfering with the enzyme lanosterol 14-α-demethylase (CYP51A1, Erg11p) encoded by the *ERG11* gene, which is involved in the transformation of lanosterol into ergosterol [6,7]. Ergosterol (ergosta-5,7,22-trien-3β-ol) is the primary sterol in the fungal cell membrane [7] and plays a major role in maintaining plasma membrane (PM) integrity and function [8,9]. Hence, its disruption has become a focus of antifungal therapies.

Unfortunately, the emergence of antifungal-resistant isolates constitutes a significant problem for treatment strategies [7]. In particular, the broad usage of azoles, such as FLC, has given rise to concerns regarding the emergence of resistance to this class of antifungal agents [3]. It is possible to distinguish several drug resistance mechanisms to azoles in *C. albicans,* including drug target alteration or overexpression, upregulation of multidrug transporters, activation of cellular stress responses or biofilm formation [10,11].

In particular, mutations in *ERG11* and amino acid substitutions in the target enzyme Erg11 lead to changes in the tertiary structure of the enzyme and subsequently alter the abilities of azole antifungals resulting in resistance in *C. albicans* [7,12]. Many studies have identified point mutations in the *ERG11* gene in azole-resistant *C. albicans* isolates. Up to now, more than 160 distinct amino acid substitutions have been reported. However, only ten of them cause FLC resistance [12]. This information suggests that the enzyme encoded by this gene is highly susceptible to structural changes. Previous reports of mutations in the *ERG11* gene have defined three hot-spot regions located within residues 105 to 165, 266 to 287, and 405 to 488, which are particularly permissive to amino acid substitutions [13]. Amino acid replacement in these hot-spot regions could correspond to conformational changes in the protein [14]. Frequently, clinical isolates of *C. albicans* reveal several amino acid substitutions as a result of long-term exposure to the antifungals. However, not all amino acid substitutions contribute equally to azole resistance [13].

*C. albicans* and lactic acid bacteria (LAB) share metabolic niches throughout the human gastrointestinal tract, such as the mouth, gut, and vagina. In vitro studies have identified various interactions between these microorganisms [15]. LAB are known to suppress filamentation, a key virulence feature of *C. albicans*, through the production of lactic acid and other metabolites [16]. The inhibitory effect of synergistic treatment of lactic acid and azoles against *C. albicans* has been observed before [17]. However, it seems interesting whether treatment with lactic acid or lactate may contribute to the development of point mutations in the *C. albicans* genome.

This study aimed to identify single nucleotide polymorphism (SNP) and to determine the effect of the identified point mutations on the amino acid sequence of the Erg11 protein in *C. albicans* cells depending on the growth phases of the microorganism, the presence of lactate as a carbon source and under the influence of FLC, the most commonly used antifungal compound.

## 2. Results

### 2.1. The Carbon Source Influences the Viability of C. albicans at Increasing Concentrations of Fluconazole

First, we examined the growth curve of *C. albicans* CAF2-1 on different carbon sources: lactate and glucose. As it turned out, the growth of *C. albicans* exhibited a diminished rate in the presence of lactate in the YPL medium compared to the control condition (YPD medium) (Figure 1A). The logarithmic growth phase was much longer, and the growth curve showed a flatter profile.

Subsequently, we monitored the growth of *C. albicans* CAF2-1 in YPD or YPL medium supplemented with increasing concentrations of FLC (in the range of 0–16 μg/mL). The obtained results expressed by optical density (OD_600_) indicated a significantly weaker growth of *C. albicans* on the YPL medium both in the presence and without FLC (Figure 1B). In turn, the viability analysis (expressed in % of viability) gave detailed information on the effect of FLC on *C. albicans’* CAF2-1 growth (Figure 1C). We observed that lactate exerted a moderate impact on *C. albicans* CAF2-1 in the presence of FLC starting from 0.5 μg/mL (82% of viability), whereas the growth inhibition on glucose started at 1.0 µg/mL of FLC (87.6% of viability). Along with the increasing concentration of the drug, the viability of the *C. albicans* CAF2-1 strain decreased on the YPD medium. The decrease in viability of the *C. albicans* CAF2-1 remained constant for cultures grown in the YPL medium, along with the increased concentration of FLC. These values fluctuated for FLC concentrations of 2–16 µg/mL but fell within the range of a 40–50% reduction in *C. albicans* growth (Appendix A).

As the concentration of the antifungal agent in the medium increased, it was possible to observe a decrease in the percentage of cell viability in both types of culture. The obtained results allow us to conclusively state that in the presence of FLC, the growth of *C. albicans* CAF2-1 strain cells in the YPL medium is visibly lower than in the YPD medium.

Based on the results of this experiment, we decided that the following concentrations of FLC would be used in further experiments: 0 (as a control); 0.25; 1; 4 and 16 µg/mL. The protocol of cultures’ preparation, cultivation and collection of biomass for point mutation testing is described in Section 4.2.2. Obtained biomass was a source of the genomic DNA isolation from *C. albicans* CAF2-1 to indicate point mutations resulting from various culture conditions.

### 2.2. FLC and Lactate Synergistically Impact the Porosity of C. albicans CAF2-1

As we have shown previously, FLC with lactate co-actively reduces ergosterol levels in PM [18]. Hence, we investigated here if the usage of lactate and FLC synergistically contributes to the permeabilization of *C. albicans* CAF2-1 PM by propidium iodide (PI) staining. As PI is a membrane-impermeable fluorescent DNA stain, the fluorescent cells indicate that FLC and lactate have an impact on the permeability across the PM. In our experiment, we observed progressive permeabilization (59.44%) of PM in *C. albicans* CAF2-1 strain cultured on YPL with the addition of FLC (Figure 2). In contrast, PI measurements on YPD with FLC indicated a 3.79% of permeabilization. Control conditions (culture on YPL and YPD) did not show any increased permeability of PM (data not shown).

### 2.3. Agarose Gel Electrophoresis, Purification of the PCR Product and Sanger Sequencing

The obtained genomic DNA was a matrix for the PCR reaction (performed according to the protocol described in Section 4.2.5), in which fragments of the *ERG11* gene were amplified. The efficiency of the PCR reaction was checked using the gel electrophoresis technique (described in Section 4.2.6). The illustration below (Figure 3) shows a pictorial result of this procedure. Expected product sizes were: 785 bp (fragment 5′′ → 3′) and 826 bp (fragment 3′′ → 5′). Next, the products were cut from the agarose gel, purified as described in Section 4.2.6 and analysed qualitatively and quantitatively. Prepared samples were sequenced by Eurofins Genomics (Ebersberg, Germany).

### 2.4. Identification of Point Mutations in ERG11 Gene

The obtained nucleotide sequences were exported to FASTA format and aligned using the EMBOSS Needle program (https://www.ebi.ac.uk/Tools/psa/emboss_needle/ accessed on 17 April 2020) using standard settings. The identified nucleotide substitutions were collected and are presented in Table 1. Among 60 types of point mutations, 11 substitutions turned out to be silent point mutations that did not result in a codon change of the amino acid (underlined). In turn, 12 substitutions (A357C, A383C, A798G, C799A, C799R, T819M, T822G, T822M, G823Y, A824G, T826G, A828T) were located in the hot-spots regions (marked in bold). Eight of them occurred on the YPD medium, whereas six were on the YPL medium.

Since only missense mutations can modify the protein conformation, the nucleotide sequences were translated into amino acid sequences using the EMBOSS Transeq tool (https://www.ebi.ac.uk/Tools/st/emboss_transeq/ accessed on 19 April 2020) and compared using the EMBOSS Needle program (https://www.ebi.ac.uk/Tools/psa/emboss_needle/ accessed on 19 April 2020). Among 46 identified amino acid substitutions listed in Table 2, one nonsense mutation (K251Stop) and six missense mutations (K119N, K128T, D225H, D225Y, R265G and W520R) should be distinguished. Interestingly, these mutations have been described in the literature up to now. Among the remaining 38 substitutions, 10 of them are located in the previously mentioned hot-spot regions (E266G, E266K, E266R, R267G, R267S, N273K, D275C, D275G, D275R and L276V) (marked in bold), 7 of them were found on the YPD medium and 4 of them on the YPL medium. In considering the presence of individual substitutions on each of the carbon sources, it was found that 14 of them occur only in strains growing on the YPD medium (D9E, I11T, F31V, L35F, A249D, Q250H, R264E, E266G, E266R, N273K, D275C, D275G, D275R and S506G). Moreover, 14 substitutions were found only in the strains growing on the YPL medium (D9Q, G10D, G10S, G10V, I11L, I11R, N12L, K259I, E260D, K262I, E266K, R267G, R267S and E517D). In turn, 11 substitutions were identified in both the strains growing on the YPD and YPL medium: I11A, I11G, I11M, I11S, I11V, N12I, N12Y, Y13N, R265K, L276V and N490K.

It should also be noted that in the group of 38 missense mutations mentioned, 26 of them were conservative mutations resulting in an amino acid change with similar properties. The remaining 12 changes were identified as non-conservative mutations resulting in amino acid substitutions with different properties than the original amino acid.

### 2.5. Identification of Amino Acid Substitutions as a Result of Aligning the Known Amino Acid Sequences of the Erg11 Protein of Selected Candida Species with the Amino Acid Sequence of the Erg11 Protein of C. albicans SC5314

Using the BLAST (https://blast.ncbi.nlm.nih.gov/Blast.cgi accessed on 24 April 2020), a local alignment of the amino acid sequence of *C. albicans* SC5314 strain (GenBank accession no.: AOW29509.1) was performed. The strain SC5314 was chosen for the alignment because it is a parental strain for CAF2-1 used in this study. The sequences were compared with sequences of different *Candida* species as well as of distinct strains of *C. albicans* available from biological databases. The results of alignment are shown in Appendix A. In the case of alignments of known amino acid sequences of the Erg11 protein of selected *Candida* species, 63 missense mutations were identified, of which 45 were found in both FLC-resistant and FLC-susceptible strains. The substitutions Y132F, N418R, D428E and K143R were identified as mutations occurring only in strains resistant to the used antimycotic. Alignment of the Erg11p amino acid sequences belonging to selected *C. albicans* strains resulted in the identification of 18 missense mutations. This group included one mutation recorded in the drug-sensitive strain, and four mutations were present in both the sequences of the sensitive and resistant isolates. Thirteen mutations were found in the sequences of the strains resistant to FLC.

## 3. Discussion

The studies indicate that *Candida* sp., specifically *C. albicans,* is responsible for most fungal infections [19]. The problem is the growing number of *Candida* strains resistant to conventional antifungal drugs such as azoles which are the drugs of choice for most *Candida* infections [20]. Various virulence mechanisms developed by *Candida* play a role in this increasing drug resistance [11]. Current data show that nearly 7% of all systemic candidiasis indicates a reduced sensitivity to the effects of azole drugs [21]. As a result, the need to develop new therapeutic methods is becoming stronger and stronger. Significant efforts have been made toward the chemical development of antifungal molecules. For instance, 1,3-thiazole derivatives having a lipophilic C4-substituent exhibited lower minimum inhibitory concentration (MIC) values compared to FLC [22]. Synthesised 14-helical β-peptides induced toxicity against *C. albicans* by forming pores in the cell membrane and disrupting intracellular organelles [23]. Modifications of enfumafungin leading to the new drug ibrexafungerp are also worth mentioning [5].

Interestingly, studies related to *C. albicans* mainly present the growth of fungal cells in the presence of glucose as the sole carbon source. The extent to which alternative carbon sources affect the innate immune response is poorly studied [24]. The available literature presents only a few reports on the impact of non-glucose carbon sources on the physiological processes of *C. albicans*. The eventual inhibitory effect of these molecules in the growth of *Candida* sp. has not been examined systematically and comparatively [17]. *C. albicans* can colonise glucose-poor niches of the human body, e.g., the intestine or the woman’s vagina, where lactate is the carbon source [24,25]. Lactate is commonly present in the human body, produced by red blood cells, the brain and muscles [26]. Interestingly, the discovery that lactate in the pathogen’s environment may inhibit their resistance to azole drugs is a promising prospect [24]. Recent studies indicate that the presence of lactate as the sole carbon source in the niche inhabited by the pathogen results in the production of a higher amount of biofilm biomass by *C. albicans*, in which blastoconidia are the dominant morphological form [25]. Importantly, biofilms composed of a mixture of yeast-form cells, pseudohyphal cells and hyphal cells surrounded by an extracellular matrix are less sensitive to antifungal drugs than planktonic cells. The change in the proportion of biofilm elements due to the presence of lactate reduces its resistance to antifungal agents [27,28]. Moreover, lactate lowers the expression of the *ERG11* gene of the ergosterol biosynthesis pathway in the early hours of *C. albicans* culture and induces delocalisation of the Cdr1 pump from the PM to vacuoles [18]. Overall, the use of lactate as a carbon source affects a large part of the genome and hence is likely to reveal many genetic differences. For instance, it has been found that cells grown in the presence of lactate display a significant reduction in the amount of mannan in the cell wall [24]. In addition, exposure to lactate induces β-glucan masking via a signalling pathway that has recruited an evolutionarily conserved receptor (Gpr1) and transcriptional factor (Crz1) from other pathways [29]. The presence of lactate induces the secretion of tartaric acid, which has the potential to modulate the TCA cycle, *C. albicans* cells adapted to lactate show a reduction of their internalisation by immune cells [30].In this study, we investigated the effect of two different carbon sources, glucose and lactate, on the survival of the *C. albicans* CAF2-1 strain in the presence of increasing concentrations of FLC. Moreover, we focused on identifying single nucleotide polymorphisms (SNP) and determining the effect of identified point mutations on the Erg11p amino acid sequence in *C. albicans* cells.

Obtained results show that the *C. albicans* CAF2-1 strain exhibit a diminished growth rate in the presence of lactate in the YPL medium compared to the control condition (YPD medium). Furthermore, cells grown on a medium with lactate as the carbon source show greater sensitivity to FLC than cultures grown on a medium with glucose. Thus, our findings confirm the information about the decrease in resistance of *C. albicans* to azoles, such as miconazole and FLC, when the sole carbon source is lactate [24,25]. It may be related to the hypothesis confirmed by Dalle et al. (2010), in which the presence of lactic acid salts in the environment of fungi increases the porosity of its cell wall [31]. Therefore, we aimed to investigate whether lactate and FLC synergistically contribute to the PM permeabilisation of *C. albicans* CAF2-1. In the case of culture in the YPL with the addition of FLC, we observed the permeabilisation of PM expressed as cell fluorescence. This effect might contribute to the entry of the antimycotic into cells. As noticed, due to the increased permeability across the cell membrane, a higher amount of the therapeutic compounds enters the pathogen’s cell and can be easily transported to the target [6]. As a result, lower concentrations of the antimycotic prove to be toxic for the fungus. Indisputably, the discovery of the interaction between lactate and resistance of *C. albicans* to azole drugs could be significant in combating fungal infections caused by strains of fungi of the genus *Candida*, which are resistant to conventional antifungal compounds. Importantly, *Lactobacillus* and as well as lactate or lactic acid, can be applied from the outside during the treatment of candidiasis [32].

The proven effect of lactate on decreasing the survival rate of *C. albicans* in the presence of FLC inspired us to analyse the point mutations of the *ERG11* gene as a result of culturing the fungus under various conditions. As a result of missense mutations causing amino acid substitutions in the protein, we observed the reduction of the degree of affinity of the active site of the Erg11 enzyme towards azoles. As a consequence, the susceptibility of the strain to a particular antifungal drug decreased. In the case of the *ERG11* gene sequence, we recorded the highest number of mutations in three hot-spot regions. Importantly, hot-spot regions are located close to the active site of the enzyme [33]. The analysis of the obtained nucleotide sequences revealed 60 types of single-point mutations. Among them, A383C, C658T, A1440G and T1470C nucleotide substitutions are already described in the literature [34]. Identification of SNP resulted in finding 45 amino acid substitutions. Up to now, six of these mutations, K119N, K128T, D225H, D225Y, R265G and W520R, are described in the literature [21,35,36]. However, none of these mutations is associated with the phenomenon of azole resistance of *C. albicans*. Hence, all identified mutations, which have not been described in the literature so far, were analysed. As amino acid substitutions may be associated with *C. albicans’* resistance to FLC, changes present only in amino acid sequences isolated from cultures showing a high viability % (≥80%) in the presence of increasing antifungal agent concentration were considered. As a result, the following hypothetical mutations may be associated with the reduced susceptibility of *C. albicans* to FLC: G10D, G10V, I11M, I11R, Y13N, F31V, L35F, A249D, Q250H, E266G, R267G, N273K, D275C, D275G, D275R. Only four indicated mutations were identified in the *ERG11* gene sequences from fungal cells grown on the YPL medium, while nine were located in those from the YPD cultures. Substitutions found in *Erg11* sequences isolated from FLC-resistant cultures and substitutions located in FLC-sensitive and FLC-resistant cultures were considered possibly unrelated to the resistance mechanism of *C. albicans*. The number of this type of mutation was 23.

Next, we aligned the amino acid sequences of the Erg11 protein derived from isolates of different *Candida* species known from the available literature with the Erg11p sequence of the *C. albicans* SC5314 strain (Appendix A). Six of the analysed sequences came from isolates sensitive to FLC activity, while the remaining nine belonged to isolates showing resistance to the antimycotic. We identified 64 missense mutations, of which only four (Y132F, N418R, D428E, K143R) were recorded in the sequences of resistant isolates. The Y132F mutation has been described before as related to the resistance of the genus *Candida* to azole drugs. The mutations were described in the isolates of *C. albicans*, *C. orthopsilosis*, *C. parapsilosis* and *C. tropicalis* [7,12,35,37,38,39]. In turn, the N418R substitution was reported and related to the resistance in *C. auris* [40]. So far, it has not been recorded in the *C. albicans*. The D428E substitution was identified in resistant (MIC ≥ 256 µg/mL) *C. tropicalis* isolate IHEM21234 [41]. In turn, K143R substitution found in *C. albicans* 10C1B1M1 had a ≥4-fold increase in FLC MIC [7]. Furthermore, the K143 substitutions were also identified in other azole-resistant clinical isolates, *Candida* spp., which demonstrated the importance of this amino acid for the proper binding of azoles to Erg11p [42,43].

Unfortunately, with the sequences of the Erg11 protein belonging to different isolates of the *C. albicans* species, it was not possible to link the amino acid substitutions that sensitise pathogen cells to FLC (Appendix A) with the results obtained by aligning the sequence of the Erg11p from *C. albicans* CAF2-1 cultured on the YPL (Table 2). We found two-point mutations in the E266 position (E266G and E266R) described before by Flowers et al. (2015). However, in their study, the E266D substitution occurred only in combination with other amino acid substitutions. It did not contribute any additional effect on azole susceptibility beyond what authors observed with the single amino acid substitution [7]. Moreover, we detected two kinds of mutations at position D225 (D225H and D225Y). These mutations have been described earlier. However, D225 is located in the F helix of Erg11p, far away from the I helix or the active centre. Thus, substitutions of D225 cannot influence the affinity between FLC and the target [44]. Similarly, the amino acid substitutions K128T and K119N reported previously were also determined not to confer resistance to FLC [7,12,36,45]. The contribution to the resistance of other mutations (R265G and W520R) is uncertain [35,44].

Nonetheless, our results suggest that the carbon source may play a role in inducing the formation of amino acid substitutions in the Erg11p. We observed a twice higher number of amino acid mutations in the YPD medium compared to the YPL medium. This result may suggest that lactate causes a lower ability of *C. albicans* to mutate. However, a lower number of lactate-induced mutations may appear due to the lower growth rate of *Candida* on the YPL compared to the YPD medium. Before, we proposed synergistic effects of lactate and FLC, where *C. albicans* was four-fold more sensitive to FLC in the presence of lactate than glucose [18]. In addition to mutations in the *ERG11* gene, a factor influencing the altered strain sensitivity of *C. albicans* to FLC may naturally be a higher or lower level of MDR proteins (Cdr1, Cdr2 and Mdr1) activity [44]. Interestingly, Derkacz et al. (2022) noticed that point mutations in the *ERG11* gene not only result in lower susceptibility to FLC. They also contribute to higher levels of ergosterol after FLC treatment or may result in chitin and β-glucan unmasking on the cell surface [46], which proves the complexity of the effects of single-point mutations in the *ERG11* gene. Moreover, single nucleotide changes in the *ERG11* gene may not affect the sensitivity of *C. albicans* strains, and only multiple substitutions may indicate a possible relation with the increase in resistance to azole drugs [45]. We should be aware that mutations can lead to the loss of function of genes. For instance, the loss of function of *ERG11* genes leads to the exchange of ergosterol for alternate sterols such as lanosterol, eburicol and 4,14-dimethyl-zymosterol in PM [47]. This mechanism may result in the acquisition of resistance of *C. albicans* to polyenes, which interact with ergosterol.

The research carried out in this study allowed us to confirm the information on the increased sensitivity of fungal cultures grown in the presence of FLC on a medium with lactic acid salts as a sole carbon source. These results support the view that the adaptation of *Candida* cells to the carbon source present in the host niches affects their pathogenicity. We have identified 15 amino acid substitutions in *C. albicans* cultures that may be part of the fungal resistance mechanism to the commonly used azole compound. These findings confirm the thesis that mutations in the *ERG11* gene are prevalent among *Candida* species, especially in clinical azole-resistant isolates, and most mutations result in appreciable changes in FLC susceptibilities [7].

## 4. Materials and Methods

### 4.1. Materials

#### 4.1.1. Strains and Growth Condition

In the study, *C. albicans* CAF2-1 (*Δura3::imm434/URA3;* parental strains: SC5314) strain was used [48]. The strain was stored at −80 °C.

The *C. albicans* strain was pre-grown in YPD medium (1% peptone, 1% yeast extract, 2% glucose) at 28 °C, 120 rpm for 24 h. Agar in a final concentration of 2% was used for medium solidification. For experiments, YPD and YPL (2% lactate, 1% yeast extract, 1% peptone) mediums were used.

#### 4.1.2. Chemicals

Chemicals and reagents used in this study were obtained from the following sources: peptone, yeast extract (manufacturer: BD; distributor: Life Technologies; Warsaw, Poland); bacteriological agar, D-glucose, Tris, propidium iodide (PI) (manufacturer: BioShop; distributor: Epro Science, Puck, Poland); ethylenediaminetetraacetic acid (EDTA), fluconazole (FLC) (Merck Life Science; Poznań, Poland), boric acid (Poch S.A., Gliwice, Poland).

### 4.2. Methods

#### 4.2.1. Determination of *C. albicans* Growth Curve and Viability in Increasing Concentration of Fluconazole

The *C. albicans* CAF2-1 was pre-grown in YPD medium (28 °C; with shaking: 120 rpm) overnight to determine growth phases. Next, cells were washed with fresh YPD or YPL medium. 100 µL of YPD or YPL medium was added to the sterile 96-well plate (Sarstedt, Nümbrecht, Germany) and inoculated with *C. albicans* to the final A_600_ = 0.05 and incubated for 24 h at 28 °C with shaking (120 rpm). The OD_600_ measurements were performed every hour using the Spark multimode microplate reader (Tecan, Männedorf, Switzerland) in three independent repetitions.

In order to determine the viability of *C. albicans* CAF2-1, a serial dilution of fluconazole (µg/mL) was prepared in YPD or YPL medium using sterile 96-well plates. Wells containing medium were inoculated with *C. albicans* CAF2-1 strain (final OD_600_ = 0.01). The volume of each well was 100 µL. Plates were incubated for 24 h at 28 °C without shaking. Next, the optical density (λ = 600 nm) was measured using Asys UVM 340 (Biogenet, Józefów, Poland). Negative controls were wells with YPD or YPL medium. The experiment was carried out in three independent replications.

#### 4.2.2. Propidium Iodide (PI) Staining of *C. albicans* Cells

In order to investigate the permeabilisation of *C. albicans* PM, propidium iodide (PI) staining was used as described before [49], with modifications. *C. albicans* cells after 24 h of culture on the YPL or the YPD with or without the addition of FLC (4 µg/mL) were washed three times with PBS (5 min, 4500 rpm) and stained with PI (0.006 mM) for 5 min, RT. Then, cells were harvested, washed three times with PBS and concentrated. The samples (*n* = 100 cells per each case) were visualised using a Zeiss Axio Imager A2 equipped with a Zeiss Axiocam 503 mono microscope camera and a Zeiss HBO100 mercury lamp (Oberkochen, Germany). The percentage of permeabilised cells was evaluated by counting PI-positive cells out of one hundred cells in three independent repetitions for each experiment.

#### 4.2.3. Cultures of *C. albicans* and Preparation of Material for DNA Isolation

For experiments, 20 mL of YPD or YPL medium (with or without FLU) was inoculated with pre-culture at starting A_600_ = 0.1 and grown for 8, 14 or 24 h (28 °C, 120 rpm). The addition of FLC to the culture was 0.25, 1, 4, or 16 µg/mL. At the end of the experiments, the cultures were transferred to 50 mL falcons or 1.5 mL Eppendorf tubes (the final OD_600_ = 1.5). Then, the cultures were centrifuged (10 min, 13 400 rpm), the supernatant was discarded, and the biomass was stored at −20 °C.

#### 4.2.4. DNA Isolation

Genomic DNA from *C. albicans* cultures was isolated using the GeneMATRIX Bacterial & Yeast Genomic DNA Purification Kit (EURx, Gdańsk, Poland) according to the manufacturer’s instructions. Purified DNA was eluted with Mili-Q water (ultrapure molecular water) at room temperature.

The amount of isolated DNA and its purity were determined by spectrophotometric analysis using the DS-11 Series Spectrophotometer / Fluorometer (DeNovix, Wilmington, DE, USA).

#### 4.2.5. Polymerase Chain Reaction (PCR)

The amplification of the *ERG11* gene sequence was performed using PCR. Due to its length (1587 bp), the sequence was amplified in two fragments (A and B) using the primers shown in Table 3. The starters were designed using the SnapGene program and the OligoAnalyzer tool and were synthesised by Genomed (Warsaw, Poland). The primers were designed to include a sequence of 20 bp downstream and 12 bp upstream (at positions 20 bp *ERG11* or +1599 bp *ERG11*, respectively) to ensure the amplification of the entire ERG11 gene sequence.

Reactions were carried out in the following reaction mixture: 12.5 µL iProof ™ HF Master Mix (BioRad, Hercules, CA, USA), 0.5 µL of each of the 5 µM primers, 2 µL of isolated genomic DNA and 9.5 µL of sterile water (total volume 25 µL). Fragment “A” of the *ERG11* gene sequence was amplified using a program called ZL-ERG11A and fragment “B” using the ZL-ERG11B program. A C100 Touch Thermal Cycler (BioRad, Hercules, CA, USA) was used to perform the PCR reaction. The reaction programs are shown in Table 4.

#### 4.2.6. Agarose Gel Electrophoresis, Purification of the PCR Product and Sanger Sequencing

In order to identify the amplified DNA fragments, 1% agarose gel electrophoresis (Bi Tools, Madrid, Spain) with the addition of the SimplySafe nucleic acid intercalator (5 μg/mL) (EURx, Gdańsk, Poland) was used. GeneRuler ™ 1 kb DNA Ladder (Thermo Fisher Scientific ™, Waltham, MA, USA) was used as the marker. Purification of PCR products from unattached nucleotides and primers was performed using the GeneMATRIX Agarose-Out DNA Purification Kit (EurX, Gdańsk, Poland) according to the manufacturer’s instructions. The samples were suspended in molecular water (Mili-Q). The purified DNA fragments were stored at −20 °C. The samples were then analysed qualitatively and quantitatively with the DS-11 Series Spectrophotometer/Fluorometer (DeNovix, Wilmington, DE, USA), and then samples were subjected to Sanger sequencing. The templates for the reaction were previously purified PCR products that were diluted according to the manufacturer’s instructions to a final concentration of 5 ng/µL with molecular water (Mili-Q). The sequencing reaction was performed by Eurofins Genomics (Ebersberg, Germany).

#### 4.2.7. The Identification of *ERG11* Point Mutations among Amino Acid Sequences of *C. albicans* Strains Available in Databases

The obtained nucleotide sequences were exported to FASTA format using the Jalview application. These were then translated into amino acid sequences using the EMBOSS Transeq tool (https://www.ebi.ac.uk/Tools/st/emboss_transeq/ accessed on 19 April 2020).

Using the BLAST (basic local alignment search tool; https://blast.ncbi.nlm.nih.gov/Blast.cgi accessed on 24 April 2020), a local alignment of the amino acid sequence of strain SC5314 (GenBank accession no.: AOW29509.1) was performed. The strain *C. albicans* SC5314 is the parental strain of CAF2-1 used in this study. The sequences were compared with sequences of different *Candida* species as well as of distinct strains of *C. albicans* available from biological databases. Nucleotide and amino acid sequences were aligned using the EMBOSS Needle program (https://www.ebi.ac.uk/Tools/psa/emboss_needle/ accessed on 19 April 2020) using standard settings. The identified nucleotide and amino acid substitutions were collected and presented in the form of a table.

#### 4.2.8. Statistical Analyses

For data analysis, statistical significance was determined using a Student’s *t*-test (binomial, unpaired). Data represent the means ± standard errors from at least three biological replicates.

## 5. Conclusions

The aim of the study was to determine the effect of point mutations of the Erg11p amino acid sequence in *C. albicans* depending on the phases of cell growth, the presence of lactate as a carbon source and the influence of FLC. As a result, *C. albicans* CAF2-1 grown on lactate as the sole carbon source showed increased sensitivity to FLC compared to growth on glucose. The synergistic treatment with lactate or glucose and FLC contributed to the development of point mutations in the *C. albicans* genome. The analysis of the *ERG11* gene derived from the CAF2-1 strain cultivated in various conditions revealed 56 nucleotide substitutions, which resulted in 38 amino acid substitutions. Ten substitutions were located in the hot-spot regions. Interestingly, almost twice a higher number of mutations in the hot-spot regions were found in the cultures grown on lactate, compared to glucose. This result may be related to the increased sensitivity of cells grown on the YPL medium to the antimycotic effect. However, there is no evidence that either mutation is responsible for the increased susceptibility of *C. albicans* CAF2-1 to FLC. Thus, our study is a good starting point for further research, such as site-directed mutagenesis, to investigate the impact of mutations on *C. albicans’* resistance/susceptibility to FLC.

## Figures and Tables

**Figure 1 pathogens-11-01289-f001:**
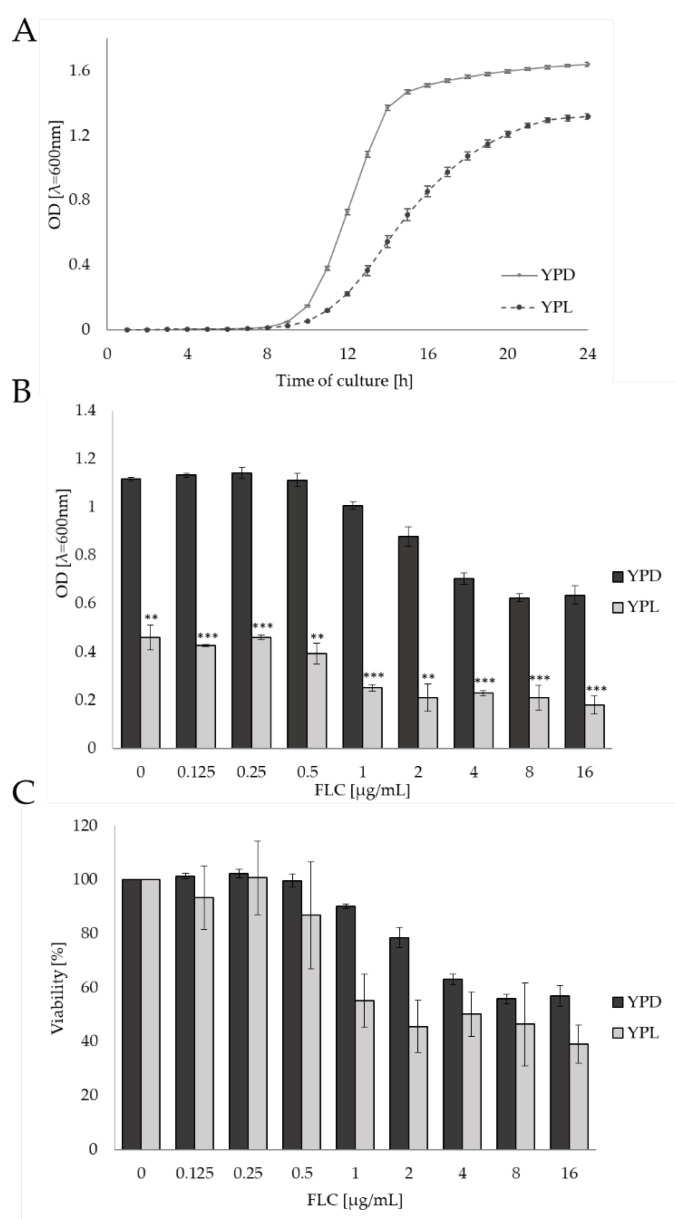
The growth of *C. albicans* CAF2-1 strain on (**A**) YPD (glucose) and YPL (lactate) medium; in increasing concentration of FLC (μg/mL) expressed as (**B**) the optical density (λ = 600 nm) and (**C**) the % of viability (**, *p* < 0.01; ***, *p* < 0.001).

**Figure 2 pathogens-11-01289-f002:**
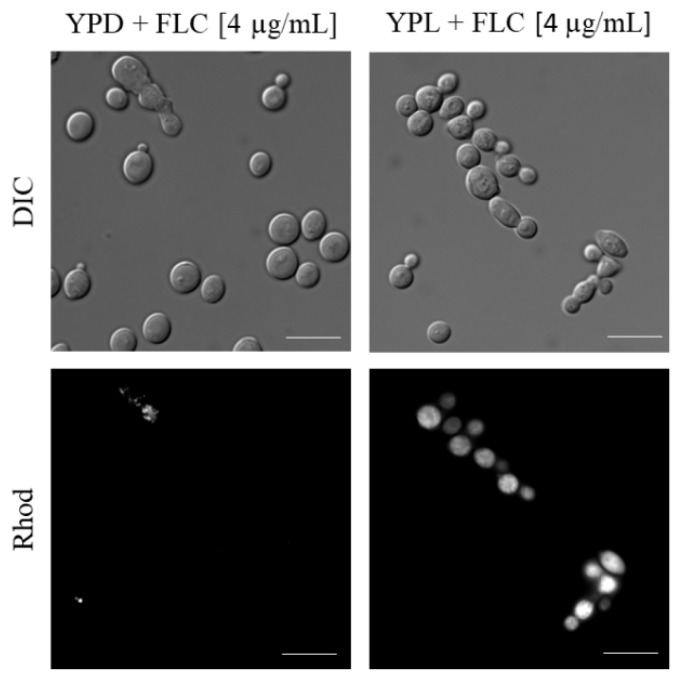
Permeabilisation of *C. albicans* CAF2-1 strain. Representative micrographs of staining with PI after 24 h of culture on YPD and YPL with FLU (4 µg/mL); scale bar: 10 μm.

**Figure 3 pathogens-11-01289-f003:**
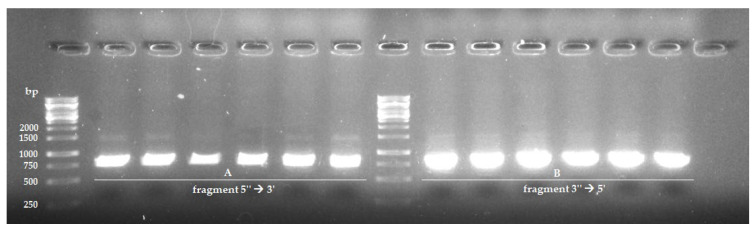
Representative electrophoretic separation presenting fragments of 5′′ → 3′ (**A**) and 3′′ → 5′ (**B**) of the *ERG11* gene. The expected sizes of the products were: 785 bp (fragment (**A**)) and 826 bp (fragment (**B**)). The size of obtained DNA fragments was compared to GeneRuler ™ 1 kb DNA Ladder.

**Table 1 pathogens-11-01289-t001:** Nucleotide substitutions of the *ERG11* gene identified in cultures of *C. albicans* CAF2-1 strain under various conditions. Hot-spot nucleotide mutations are marked in bold; previously identified mutations are marked in italics; mutations that did not result in an amino acid substitution are underlined.

			Concentration of Fluconazole [µg/mL]				
			0	0.25	1	4	16
Carbon source	Glucose	8 h	C30T, **T819M, T826G, A828T**	C30T, A31G, T37A, **T819M, T826G, A828T**	A105T, C746A, A751T, **T826G, A828T,** T1558C	A35T, T37A, A797G, **T826G, A828T,** T1558C	C30A, A31G, T37A, ***A383C*, T826G, A828T,** *T1470C*, T1558C
		14 h	C30T, A35T, T37A, A790G, G791A, G794A	A35T, T37A, A657G, T768A, **T826G, A828T,** *T1470C*	A35T, A105T, C746A, **T826G, A828T**	T33G, A35T, *A383C*, A750T, **T826G, A828T,** *T1470C*, T1558C	T27G, C30T, A34T, **A357C, *A383C*, T826G, A828T,** *T1470C*, T1558C
		24 h	C30W, A31G, T32S, A90G, T92G, **T826G, A828T**	C30W, A31G, T32S, **T822M, G823Y, A824G, T826G, A828T**	T33G, A35T, T37A, G673C, **T826G, A828T,** *A1440G*, A1516G	A31G, T32S, G791A, G796A, A797G, **T826G, A828T,** A1516G	C30T, T32S, A35T, **T826G, A828T,** T1470A, A1516G
	Lactate	8 h	A31G, T32S, **T826G, A828T**	T32G, T33S, A34T, T37A, **C799R, T826G, A828T**	C30W, A31G, T32S, **T826G, A828T**	G29W, A35T, T37A, **T826G, A828T,** T1470A	C30A, A31S, T36A, **T826G, A828T,** *T1470C*
		14 h	T33G, A34T, G791A, A792G, G794A, **T826G, A828T**	T33G, A34T, **T826G, A828T,** *T1470C*	A34T, G673T, **T826G, A828T**	C30T, A34T, *A383C*, **T826G, A828T,** *T1470C*	T26G, C30T, *A383C*, A776T, A790G, **T826G, A828T**
		24 h	A31G, T32S, A790G, **T826G, A828T,** *A1440G*, *T1470C*	*C658T*, A776T, G794A, **T822G, T826G, A828T**	A31G, T33G, A793G, **T826G, A828T**	A31G, T32S, *A383C*, A780C, A785T, **T826G, A828T**	G28A, A35T, T36A, *A383C*, A780C, G796A, **A798G, C799A, T826G, A828T,** A1551T

**Table 2 pathogens-11-01289-t002:** Amino acid substitutions of the *ERG11* gene identified in cultures of *C. albicans* CAF2-1 strain under various conditions. Hot-spot amino acid mutations are marked in bold; previously identified mutations are marked in italics; mutations that did not result in an amino acid substitution are underlined.

			Concentration of Fluconazole [µg/mL]				
			0	0.25	1	4	16
Carbon source	Glucose	8 h	**N273K, L276V**	I11V, Y13N, **N273K, L276V**	L35F, A249D, K251STOP, **L276V**, *W520R*	N12I, Y13N, **E266G, L276V**, *W520R*	I11V, Y13N, ***K128T*, D275G, L276V**, *W520R*
		14 h	N12I, Y13N, R264E, R265K	N12I, Y13N, **L276V**	N12I, L35F, A249D, **L276V**	I11M, N12I, ***K128T***, Q250H, **L276V**, *W520R*	D9E, N12Y, ***K119N*, *K128T*, L276V**, *W520R*
		24 h	I11A/I11G, F31V, **L276V**	I11A/I11G, **D275R/D275C, L276V**	I11M, N12I, Y13N, *D225H*, **L276V**, S506G	I11A/I11G, R265K, **E266R, L276V**, S506G	I11T/I11S, N12I, **L276V**, N490K, S506G
	Lactate	8 h	I11A/I11G, **L276V**	I11S/I11R, N12Y, Y13N, **R267S/R267G, L276V**	I11A/I11G, **L276V**	G10D/G10V, N12I, Y13N, **L276V,** N490K	I11L/I11V, N12L, **L276V**
		14 h	I11M, N12Y, *R265G*, R265K, **L276V**	I11M, N12Y, **L276V**	N12Y, *D225Y*, **L276V**	N12Y, ***K128T*, L276V**	D9Q, N12Y, ***K128T***, K259I, *R265G*, **L276V**
		24 h	I11A/I11G, *R265G*, **L276V**	K259I, *R265G*, **L276V**	I11V, *R265G*, **L276V**	I11A/I11G, ***K128T***, E260D, K262I, **L276V**	G10S, N12I, ***K128T***, E260D, **E266K, R267S**, **L276V**, E517D

**Table 3 pathogens-11-01289-t003:** Primers used to amplify *ERG11* gene fragments.

Fragment	Name	Sequence	Product Size
A	ERG11-FA	5′- ATGGCTATTGTTGAAACTGTC-3′	785 bp
	ERG11-RA	5′- CGTTCTCTTCTCAGTTTAATTTC-3′	
B	ERG11-FB	5′- GAAGAGAACGTGGTGATATTGATC-3′	826 bp
	ERG11-RB	5′- CACTGAATCGAAAGAAAGTTGCC-3′	

**Table 4 pathogens-11-01289-t004:** Description of the ZL-ERG11A and ZL-ERG11B programs used for the amplification of “A” and “B” fragments of the *ERG11* gene.

Step	Program				Cycles
	ZL-ERG11A		ZL-ERG11B		
	Time	Temperature	Time	Temperature	
Initial denaturation	5 min	98 °C	5 min	98 °C	-
Denaturation	30 s	98 °C	28 s	98 °C	35
Annealing	30 s	51 °C	29 s	55 °C	
Extension	30 s	72 °C	30 s	72 °C	
Final extension	5 min	72 °C	5 min	72 °C	-
	∞	4 °C	∞	4 °C	-

## Data Availability

Not applicable.

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
