# Peer review of "The Role of ERG11 Point Mutations in the Resistance of Candida albicans to Fluconazole in the Presence of Lactate"

_pathogens, 2022, doi:10.3390/pathogens11111289_

Round 1

Reviewer 1 Report

It is an original article that can be of high impact for the scientific community since it has the scientific rigor to be published.  

Author Response

Thank you for your comment. We appreciate the time you spend reviewing our publication.

Reviewer 2 Report

The article entitled "The role of ERG11 point mutations in the resistance of Candida albicans to fluconazole in the presence of lactate" by Urbanek AK. Et al., describes the effect of different carbon sources in acquiring resistance against azole drug FLC. The article in its current form does not have sound results to support the conclusions made. There are many drawbacks in the study to make the given conclusions. The language needs to improve and needs to be relooked at throughout the manuscript.

Some of the concerns include:

1.     First half of the manuscript does not seem to be fully novel as the use of lactate as a Carbon source and associated results seem to be known earlier. The references given in the manuscript like 20-23 and others in the discussion also suggest that the use of lactate a as carbon source and associated decrease in drug resistance have been suggested earlier as well.

2.     Many sentences are either not clear or difficult to understand to the reader; like in section 2.2 sentence….” Along with the increasing concentration….decreases on average every 12 percentage points (pp)” what do the authors mean by that?

3.     In Figures1 B & C, the percentage reduction in growth on FLC treatment in presence of YPD and YPL needs to be calculated to check the difference if any related to growth in presence of FLC. CAF2-1 doesn’t seem to get more susceptible in presence of FLC at higher drug conc . in YPL media.

4.     Reduction in PM ergosterol levels may make cells more resistant to polyenes. That aspect needs to be evaluated properly.

5.     Either use YPD or YPG in the manuscript. Both confuse the reader.

6.     There are mutations at 0 ug drug concentration. There are mutations both in YPD and YPL media. You are not clear whether to relate them to the resistance or susceptibility of CAF2-1 in your manuscript. why is growth less in ypd media when mutations are more in ypl media.

7.     How can lactate C-source increase biofilm biomass but decrease the overall growth of CAF2-1 cells?

8.     How can you relate the mutations to drug resistance or susceptibility? Can you create the mutations in WT cells and show a similar impact on drug susceptibility patterns? That would give some weight to your results.

9.     There can be an impact on other genes/pathways in presence of lactate as a carbon source. One cannot focus on the gene of choice. Until the mutations replicated in WT cells can have similar effects. There can be other genes having a role in drug resistance etc which can show impactful mutations

10.  Authors are not clear with their own conclusions- what to infer from the study.

Author Response

The article entitled "The role of ERG11 point mutations in the resistance of Candida albicans to fluconazole in the presence of lactate" by Urbanek AK. Et al., describes the effect of different carbon sources in acquiring resistance against the azole drug FLC. The article in its current form does not have sound results to support the conclusions made. There are many drawbacks in the study to make the given conclusions. The language needs to improve and needs to be relooked at throughout the manuscript.

Thank you for your valuable comments. We appreciate the time you spend reviewing our publication. We hope you will be satisfied with our answers and improved language.

Point 1: First half of the manuscript does not seem to be fully novel as the use of lactate as a Carbon source and associated results seem to be known earlier. The references given in the manuscript like 20-23 and others in the discussion also suggest that the use of lactate a as carbon source and associated decrease in drug resistance have been suggested earlier as well.

Response 1: The results from the first half of the manuscript were the starting point for further research connected to the identification of mutations. We wanted to confirm earlier observations and we did it. The novelty in this part of the manuscript is the determination of permeabilization of C. albicans CAF2-1 plasma membrane (PM) by propidium iodide (PI) staining in cells grown on lactate and treated with FLC. The second part is focused on the identification of arisen mutations in the Erg11 gene of C. albicans. As we mention in the Discussion, most of the found mutations have been previously described. They have appeared in azole-resistant clinical strains, which only confirms that the mutations that were found in this study may play a role in antimycotic resistance. However, the rest of the mutations (not described before) may be a subject for site-directed mutagenesis to investigate the impact on FLC resistance/susceptibility. Thus, we think that identified mutations in this study could be a good starting point for further research.

Point 2: Many sentences are either not clear or difficult to understand to the reader; like in section 2.2 sentence….” Along with the increasing concentration….decreases on average every 12 percentage points (pp)” what do the authors mean by that?

Response 2: The paraphrased sentence was included in section 2.1. We agree that the construction of this sentence could be unintelligible, so we edited it. Furthermore, we revised the manuscript once again and did our best to make it easier to understand.

Point 3: In Figures1 B & C, the percentage reduction in growth on FLC treatment in presence of YPD and YPL needs to be calculated to check the difference if any related to growth in presence of FLC. CAF2-1 doesn’t seem to get more susceptible in presence of FLC at higher drug conc . in YPL media.

Response 3: In section 2.1 we underlined that lactate exerts a moderate effect against C. albicans CAF2-1 in presence of FLC starting from 0.5 μg/mL. The decrease in viability remains constant along with the increased concentration of FLC. Values fluctuate for FLC concentrations of 2-16 µg/mL, however, fall within the range of a 40-50% reduction in C. albicans growth. We calculated the percentage reduction in growth and presented it in the table attached to the Supplementary Material as Table S1.

Point 4: Reduction in PM ergosterol levels may make cells more resistant to polyenes. That aspect needs to be evaluated properly.

Response 4: Thank you for your suggestion. We added a relevant fragment to the discussion (lines 362-366).
Point 5: Either use YPD or YPG in the manuscript. Both confuse the reader.

Response 5: Thank you for your comment. We replaced “YPG” with “YPD” in the legend of Figure 1.

Point 6: There are mutations at 0 ug drug concentration. There are mutations both in YPD and YPL media. You are not clear whether to relate them to the resistance or susceptibility of CAF2-1 in your manuscript. why is growth less in ypd media when mutations are more in ypl media.

Response 6: Indeed, there are mutations at a concentration of 0 µg/mL FLC. We do not exclude that they may be spontaneous mutations. However, in our opinion, there are fewer of these mutations compared to other concentrations. Anyway, as we indicated in the discussion, these mutations (located in FLC-sensitive and FLC-resistant cultures) were considered possibly unrelated to the resistance mechanism of C. albicans. Moreover, we showed that some of the mutations were identified before and their function is not connected with FLC resistance or is unclear (lines 314-345).

In discussion, we wrote: “We observed a twice higher number of amino acid mutations in the YPD medium compared to the YPL medium. This result may suggest that lactate causes a lower ability of C. albicans to mutate. However, a lower number of lactate-induced mutations may appear due to the lower growth rate of Candida on the YPL compared to the YPD medium.”

Point 7: How can lactate C-source increase biofilm biomass but decrease the overall growth of CAF2-1 cells?

Response 7: In the sentence in lines 241-244, we wanted to underline that indeed lactate increases biofilm biomass, but the dominant morphological form is yeast-form cells, which can divide faster than pseudohyphal cells and hyphal cells but still may be protected by other compounds of biofilm such as a matrix. It means that lactate influences the structure of the biofilm. We revised sentences to make them more precise and understandable for readers. It should be noted that the biofilm growth and the planktonic growth of cells differ from each other. There are metabolic and other differences behind this. For example, some metabolic pathways that are down-regulated in cells during stationary growth (compared with exponentially growing cells) have been reported to be further downregulated in cells of biofilms.

Point 8: How can you relate the mutations to drug resistance or susceptibility? Can you create the mutations in WT cells and show a similar impact on drug susceptibility patterns? That would give some weight to your results.

Response 8: We underlined that “single nucleotide changes in the ERG11 gene may not affect the sensitivity of C. albicans strains and only multiple substitutions may indicate a possible relation with the increase in resistance to azole drugs”, as supported by the literature data. In the discussion of the results, we tried to discuss in detail the influence of the detected mutations on the basis of the latest available literature. Therefore, following your train of thought, it would be necessary to create both single and double or even triple mutants. Identified mutations in our research could be a good starting point for further research focused on site-directed mutagenesis to investigate the impact of mutations on FLC resistance/susceptibility.

Point 9: There can be an impact on other genes/pathways in presence of lactate as a carbon source. One cannot focus on the gene of choice. Until the mutations replicated in WT cells can have similar effects. There can be other genes having a role in drug resistance etc which can show impactful mutations

Response 9: We agree with your statement that lactate as a carbon source can impact other genes/pathways. In this study, we focused on mutations in the Erg11 gene because cytochrome P450 lanosterol 14α-demethylase encoded by this gene is a target for fluconazole, the antifungal agent used in our study. However, in the literature, there are several reports of the effects of lactate on C. albicans cells. We have added some examples to the discussion section (lines 251-258).

Point 10: Authors are not clear with their own conclusions- what to infer from the study.

Response 10: Thank you for this comment. We improved our conclusions.

Reviewer 3 Report

In the paper entitled "The role of ERG11 point mutations in the resistance of Candida albicans to fluconazole in the presence of lactate", the authors evaluate the influence of lactate and glucose on C. albicans' resistance towards fluconazole. The idea of this research is a good one, considering the high incidence of multi-drug resistant fungal infections. Also, the paper is well organized and the information is clearly presented.

However, it still needs a minor revision before acceptance:

- Minor English language editing by an expert (there are some singular/plural errors);

- I advice the authors to replace the word "antibiotic"....with "antifungal" or "antimicrobial" ...even if it is not wrong, we mostly use the term "antibiotic"  when referring to an antibacterial agent;

- all abbreviations should be defined when first used within the text;

- page 1 - line 15: replace "drugs that are more efficient and tolerant" with "drugs that are more efficient and with a better toxicological profile" (tolerance belongs to the pacient, not to the drug);

- the authors should mention ibrexafungerp, the newest antifungal approved to treat vulvovaginal candidiasis;

- References: more than 35% of the references cited are more than 10 years old; I advise the authors to insert a more up to date bibliography (eg. Pricopie, A.-I.; Focșan, M.; Ionuț, I.; Marc, G.; Vlase, L.; Găină, L.-I.; Vodnar, D.C.; Simon, E.; Barta, G.; Pîrnău, A.; Oniga, O. Novel 2,4-Disubstituted-1,3-Thiazole Derivatives: Synthesis, Anti-Candida Activity Evaluation and Interaction with Bovine Serum Albumine. Molecules 202025, 1079. https://doi.org/10.3390/molecules25051079' etc.)

Author Response

In the paper entitled "The role of ERG11 point mutations in the resistance of Candida albicans to fluconazole in the presence of lactate", the authors evaluate the influence of lactate and glucose on C. albicans' resistance towards fluconazole. The idea of this research is a good one, considering the high incidence of multi-drug resistant fungal infections. Also, the paper is well organized and the information is clearly presented.

Response: Thank you for all your comments. We hope that our answers are accurate to the raised issues. We appreciate the time you spend reviewing our publication.

However, it still needs a minor revision before acceptance

Point 1: Minor English language editing by an expert (there are some singular/plural errors).

Response 1: We did our best to improve the language. We found errors in the text and corrected them.

Point 2: I advise the authors to replace the word "antibiotic"....with "antifungal" or "antimicrobial" ...even if it is not wrong, we mostly use the term "antibiotic"  when referring to an antibacterial agent.

Response 2: According to your suggestion, we replaced the word “antibiotic” with either “antifungal agent” or “antimicrobial agents” in lines: 32, 101 and 305.

Point 3: All abbreviations should be defined when first used within the text.

Response 3: We have added ‘FLC’ in line 33, the place where fluconazole is mentioned in the main text for the first time. We added “plasma membrane” in line 42 and then used only “PM”.

Point 4: Page 1 - line 15: replace "drugs that are more efficient and tolerant" with "drugs that are more efficient and with a better toxicological profile" (tolerance belongs to the patient, not to the drug).

Response 4: In line 15 we changed the wording according to your suggestion.

Point 5: The authors should mention ibrexafungerp, the newest antifungal approved to treat vulvovaginal candidiasis.

Response 5: W mentioned ibrexafungerp in the Introduction (lines 35 – 36) and Discussion sections (lines 227-228).  

Point 6: References: more than 35% of the references cited are more than 10 years old; I advise the authors to insert a more up to date bibliography (eg. Pricopie, A.-I.; Focșan, M.; Ionuț, I.; Marc, G.; Vlase, L.; Găină, L.-I.; Vodnar, D.C.; Simon, E.; Barta, G.; Pîrnău, A.; Oniga, O. Novel 2,4-Disubstituted-1,3-Thiazole Derivatives: Synthesis, Anti-Candida Activity Evaluation and Interaction with Bovine Serum Albumine. Molecules 2020, 25, 1079. https://doi.org/10.3390/molecules25051079' etc.)

Response 6: We agree with your opinion. More than 35% of the references cited are more than 10 years old, so we insert more recent entries (10 positions), also the one suggested by you (Quindós 2022; Jallow 2021, Pricopie 2020, Lohse 2018, Winter 2016, Lee 2019, Ballou 2016, Oliveira-Pacheco 2018, Carolus 2020).